# Solubilization of Nile Red in Micelles and Protomicelles of Sodium Dodecyl Sulfate

**DOI:** 10.3390/molecules27227667

**Published:** 2022-11-08

**Authors:** Anatoly I. Rusanov, Tamara G. Movchan, Elena V. Plotnikova

**Affiliations:** 1Frumkin Institute of Physical Chemistry and Electrochemistry, Russian Academy of Sciences, 119071 Moscow, Russia; 2Mendeleev Center, St. Petersburg State University, 199034 St. Petersburg, Russia

**Keywords:** sodium dodecyl sulfate, nile red, solubilization, adsorption, surfactants, hydrophobic effect, micelles, protomicelles, nano-adsorbent, molecular aggregation

## Abstract

A spectrophotometric study of the solubilization and aggregation of the Nile red dye (NR) in premicellar and micellar aqueous solutions of sodium dodecyl sulfate (SDS) was carried out. The experiments were conducted both with saturated solutions of NR under conditions of thermodynamic equilibrium of the solution with a dye precipitate, and at a constant concentration of NR in a homogeneous solution. In the first case, it was proved theoretically and verified experimentally that with an increase in the SDS concentration, the NR concentration always increases, and at the limit of low concentrations, the dependence is linear. In both cases, the concentration of NR dimers as a function passes through a maximum in the premicellar region. There are no dimers in the micellar region. The extinction coefficients of NR monomers in SDS solutions were determined both below and above the critical micelle concentration (CMC) of SDS. A solubilization curve with branches for the premicellar and micellar regions was constructed, the intersection of which was used to find the CMC value in the system under study. The state of deep supersaturation of the NR solution in the metastable state upon dilution of the micellar system with water was studied. It was found that, in addition to dimers, molecular aggregates of higher orders were also formed.

## 1. Introduction

Solubilization is a well-known phenomenon in chemistry, when the solubility of a certain substance (solubilizate) in a given solvent increases when another substance (solubilizer) is added to the solution. Solubility is understood as the concentration of the solubilizate in a solution that is in equilibrium with the pure phase of the solubilizate (for example, its crystalline precipitate). Surfactants are often used as solubilizers, especially colloidal surfactants, which form micelles when the critical micelle concentration (CMC) is reached. Inside the micelle, an environment is created that promotes the dissolution of the solubilizate, which it penetrates. This is exactly the mechanism of the washing action of the micellar solution. The main characteristic of solubilization is the solubilization capacity of micelles:(1)$$s\equiv \frac{z}{n}$$
where *z* is the number of solubilizate molecules in a micelle and *n* is the number of surfactant molecules (surface active ions if any) in a micelle (the aggregation number). Note that the aggregation number of micelles with a solubilizate can be significantly higher than in the absence of solubilizate [1].

A special case is the solubilization of single large molecules, which in a surfactant solution can play the role of a nano-adsorbent. The latter differs from conventional (macroscopic) adsorbents in that, due to the relative smallness of its particles, it becomes an equal component of the solution, having the concentration, chemical potential, and other attributes of a thermodynamic component of the system [2]. Covered with adsorbed surfactant molecules, nano-adsorbent particles become similar to micelles with solubilizate. Such micellar particles are called protomicelles [2,3]. Protomicelles do not have CMC and, in principle, can be formed even with non-colloidal surfactants. For colloidal surfactants, protomicelles can appear at concentrations much lower than the CMC [4,5].

The first experiments with nano-adsorbents and protomicelles were carried out using nanotubes and phthalocyanine dye molecules as solubilization nuclei (see Ref. [2] for the full literature). To expand the database, we now turn to another dye, Nile red (NR). Its chemical name is 9-diethylamino-5-benzo[a]phenoxazinone, and the molecules have a flat structure (Figure 1).

NR is highly sensitive to the polarity of its microenvironment and exhibits intense fluorescence [6,7,8,9,10]. NR is a hydrophobic substance with very low solubility in water (according to Ref. [6], less than 1 mg/L, i.e., less than 3 μM, and according to our data, about 2 μM [11]). Like phthalocyanines, NR has a planar molecular structure, which allows the molecules to stick together cofacially in an aqueous medium under the action of mainly hydrophobic interactions. This effect is more pronounced for phthalocyanines, whose molecules are much larger and the hydrophobic interactions are much stronger than for NR. Since the functional (chromophoric and fluorophoric) properties of dyes are largely lost in such an aggregation process, the problem of monomerization arises for them. For NR, there is no such problem; in fresh aqueous solutions, NR remains predominantly in the monomeric form. However, later, the aggregation of NR molecules reveals itself. If, for example, an aqueous solution of NR is prepared by mixing a small amount of a concentrated solution of NR in methanol with water, then at the initial moment a non-equilibrium supersaturated solution of NR monomers in water is formed. Here, aggregation is inevitable and easily observed, and the study of the kinetics of the process indicates a second-order reaction, i.e., the formation of dimers [7]. However, dimers are also formed under normal conditions, but more slowly than the dissolution of NR [11]. We return to this later. In the meantime, we can conclude that many cases in the literature of spectra with the presence of only NR monomers refer to insufficient equilibrium systems and require further study.

Classical sodium dodecyl sulfate (SDS), which has been successfully used in studies with phthalocyanines [4,12,13], was chosen as the surfactant in this work. SDS has also been used mixed with NR in aqueous solutions [9,14,15]. This work continues the study of the influence of the concentration and structure of surfactants on the aggregative state of dyes in the aqueous environment. The work is aimed at studying the effect of SDS on the solubility of NR by a spectrophotometric method with a more detailed study of the premicellar region of the surfactant both at a constant and varying dye concentration. The acquisition of new data will contribute to the development of the theory of solubilization with the help of surfactants, as well as the successful application of NR in the field of nanoscience and nanotechnology.

## 2. Materials and Methods

### 2.1. Reagents and Instruments

SDS and NR preparations from Acros Organics with the content of the main substance of 99.8% and 99.0%, respectively, were used without further purification. Water was taken in the form of a tridistillate with a specific electrical conductivity not higher than 4 × 10^–4^ mS/m at 25 °C. The absorption spectra were recorded on a UV spectrophotometer (Unico 2800, United Products & Instruments, Dayton, NJ, USA) in the wavelength range of 190–900 nm using standard quartz cuvettes with an optical path length of 1 cm.

The number of substances in the system under study could increase by one if we count protonation: the chemical reaction of the addition of a proton to the cyclic nitrogen atom in NR. Protonation occurs in an acidic environment, but the distillate of water, capturing carbon dioxide from the air during distillation, only just becomes an acidic environment. However, such a medium is still slightly acidic, and a strong acid is desirable for protonation. For example, NR protonation can be easily observed in methanol or water–methanol mixture in the presence of sulfuric acid [16]. Serious changes begin at an H_2_SO_4_ concentration of 0.05 M. In this case, the NR absorption spectrum also changes: the peaks are localized in the long-wavelength region above 600 nm. Spectra of this type were not found in this work. Therefore, we can conclude that we were dealing with an unprotonated or weakly protonated form of NR.

### 2.2. Methodology

An important detail of the technique is the preparation of an aqueous solution of NR. Since the solubility of NR in water is very low (about 2 μM [11]), the dissolution is also slow, however, NR dissolves well in organic liquids. Therefore, a typical way to reduce the waiting time is to add ready-made solutions of NR in organic solvents to water, methanol [7], ethanol [8,14,15], tetrahydrofuran [9], etc. However, we consider this method unacceptable for a number of reasons. First, the system becomes contaminated with an admixture of organic matter. In some cases, it can be eliminated (for example, methanol does not form an azeotrope with water and can be separated from water by distillation), but this creates additional procedures that distract from the main goal of the study. If we neglect the resulting impurity, there will be errors in the measurements. Water itself is a complex structured fluid and therefore sensitive to impurities in the system. Here is an example: the maximum optical absorption of NR monomers in water with 2% methanol corresponds to a wavelength of λ_max_ = 580 nm [7], while for NR monomers in pure water, λ_max_ = 593 nm [9,11]. Second, such a procedure leads, as a rule, to nonequilibrium states and, consequently, to relaxation processes, which complicates the study. Third, as a consequence, in this case too, waiting is necessary, so that the task of avoiding waiting may not be fulfilled or only partially fulfilled.

For these reasons, we did not use NR auxiliary solutions and worked in the old-fashioned way, paying with our own time. As an example, we present Figure 2, illustrating the kinetics of establishing spectroscopic data for a pure aqueous solution of NR. λ_max_ = 593 nm corresponds to NR monomers, and we see that when moving from “a” to “b” (i.e., in 6 days), the peak of optical density *A* of NR monomers, and, consequently, their concentration in solution, doubles. This means that NR is being dissolved in water in the monomeric form, but dimers have not yet appeared. After 16 days, dimers are already visible at λ = 543 nm (Figure 2c). The absorption spectrum of NC measured after 2 months, practically coincides with Figure 2c.

At the same time, the concentration of monomers decreases, since dimers and larger aggregates are formed at the expense of monomers. Here we are once again faced with the fact that the NR aggregation process is slower than the dissolution process. In this case, the total concentration of NR taken is only slightly higher than its solubility, and therefore the dissolution occurs at the lowest rate for recognizing its different stages. The larger the contact surface with the solution of the solid part of NR, the faster is the dissolution. Obviously, to speed up working preparations, you only need to take more reagent (with proper dispersion). The presence of surfactants is also a factor accelerating the achievement of equilibrium. In this work, saturated NR solutions were prepared using dispersions with a gross concentration of 440 μM, when the equilibrium was actually reached in two days. To guarantee and control the finished sample, it was kept for five days. The experiment was carried out in two versions: with solutions saturated with respect to NR (and, therefore, at a constant NR chemical potential), which were prepared as described above, and, alternatively, with a constant concentration of NR. In the first variant, the constancy of the chemical potential NR was ensured by the presence of a precipitate of NR. With this case, we begin the discussion of the results.

## 3. Results and Discussion

### 3.1. SDS in Saturated Aqueous NR Solution

The thermodynamic theory of the effect of surfactants on the concentration of a solubilizate (nano-adsorbent) in its saturated solution (when the chemical potential of the solubilizate is constant) was formulated earlier [2]. One of the formulas is
(2)$$c_{2}=c_{2}^{0}\exp\left(\frac{un}{kT}\right)$$
where $c_{2}$ is the NR concentration in our case, $c_{2}^{0}$ is the value of the NR concentration in the absence of SDS, *u* is the work of detachment of one DS^–^ ion from the nano-adsorbent surface, *n* is the number of adsorbed DS^–^ ions per NR molecule (protomicelle aggregation number), *kT* has the usual meaning. Equation (2) shows that the NR concentration increases with the number of adsorbed ions, but the latter, whatever the adsorption equation, always increases with the adsorbate concentration $c_{1}$ in solution. For small $c_{1}$, this increase is linear:(3)$$n=K_{H}c_{1},$$
where *K_H_* is Henry’s constant. In the limit of small *n*, the work *u* becomes a constant, and the dependence on in the leading term of the expansion of the exponent also becomes linear:(4)$$c_{2}=c_{2}^{0}(1+k_{1}c_{1}+\cdot \cdot \cdot ),$$
where $k_{1}\equiv uK_{H}/kT.$ Thus, Equation (2), as well as Equation (4) following from it, predicts an increase in the concentration of NR in solution with the addition of SDS.

This pattern is well illustrated in Figure 3, which shows the dependence of the NR spectra on the concentration of SDS. Region “a” corresponds to concentrations below the SDS CMC (about 8.2 mM in pure water), and “b” corresponds to concentrations above the CMC.

In both cases, there is an increase in the optical density *A* and, consequently, the concentration of NR with an increase in the SDS concentration. Interestingly, on curve *3* in Figure 3a, a peak appears at λ_max_ within the range 540–545 nm, which can be attributed to NR dimers (with a characteristic hypsochromic shift with respect to the main peak for monomers), while no signs of the existence of dimers are seen on other curves. This can be explained as follows. For two monomers to stick together to form a dimer, two conditions are necessary. First, this is the presence of a force that ensures the adhesion of monomers. We have such a force. This is a hydrophobic interaction. Second, for monomers to stick together, they must often collide with each other, and this depends on their concentration. According to the mass action law, the concentration of dimers (D is the chemical symbol of a dimer) is related to the concentration of monomers (M is the chemical symbol of a monomer) by the relation
(5)$$c_{D}=K_{d}c_{M}^{2}$$
where *K_d_* is the dimerization constant. Equation (5) shows that when $c_{M}$ is small, $c_{D}$ can be vanishingly small, which manifests itself in curves *1* and *2* in Figure 3a. In other words, there are dimers, but their concentration is so low that the device does not detect it. With the addition of SDS to the solution, both concentrations in Equation (5) increase, and dimers are detected spectroscopically in curve *3*. When there are many SDS ions in the solution (approaching the CMC) and the solubilization nuclei (NR monomers and dimers) become adsorption saturated, then in search of adsorption sites, SDS ions begin to break up the dimers. This is the known monomerizing function of surfactants in systems with dyes [17]. As a result, the dimers disappear, as can be seen from curve *4* and subsequent curves, which already refer to the micellar system. As follows from Figure 3b, with an increase in the SDS content, the optical density *A* in the spectra at λ_max_ = 579 nm increases, which indicates an increase in the amount of NR monomers solubilized in surfactant micelles.

It follows from the above that, as SDS is added, the concentration of NR dimers in the pri-micellar solution should pass through a maximum. This conclusion can also be supported by considerations related to the mass action law, Equation (5). The fact is that the dimerization constant depends on the adhesion strength of the monomers and, consequently, on the SDS concentration that affects it. The stronger the adhesion, the greater the value of *K_d_*. With an increase in $c_{1},$ the number of adsorbed SDS ions on the surface of the NR molecule increases too. This makes the surface less hydrophobic and increases the solubility of the NR monomers in water, but at the same time reduces the strength of the aggregation of monomers into dimers. The latter is provided by the hydrophobic effect, which naturally decreases as the hydrophobicity of the surfaces decreases. From a mathematical point of view, we have a confrontation between two factors on the right side of Equation (5): a decrease in *K_d_* and an increase in $c_{M}^{2}$ with the increase in $c_{1}.$ Usually, such a situation leads to the appearance of an extremum. Since, according to the physical meaning of $c_{D}$, it cannot be a negative value and has zero values at the edges, obviously, we can only talk about the maximum of $c_{D}.$

Figure 4 shows the experimental dependence *A*(*c*_1_) of the optical density on the SDS concentration, plotted from the values of *A* in the absorption maxima of NR monomers in their spectra (Figure 3). The initial section (curve *1* in Figure 4) of the dependence reflects the fact that SDS has a weak effect on the dissolution of NR in the concentration region where there are no micelles yet and only protomicelles are being formed. Gradual filling of the surface of NR molecules with SDS ions increases their solubility and leads to an increase in the concentration of NR monomers in solution, albeit slightly. In parallel, the optical density of the solution also increases. The break in the curve and the sharp increase in the increase in optical density (curve *2*) are due to the appearance and increase in the number of SDS micelles capable of solubilizing NR. The break point corresponds to the CMC. The most accurate way to find it is as follows. Curve *1* and curve *2* (we take the first 4 points) are well approximated by straight lines (R^2^ is the coefficient of determination)
(6)$$A=0.002+0.0045c_{1}\;(R^{2}=0.9626,\;c_{1}\;<\;\mathrm{CMC}),$$
(7)$$A=0.1412c_{1}-1.0178\;(R^{2}=0.9909,\;c_{1}\;>\;\mathrm{CMC}),$$
where $c_{1}$ is expressed in mM. The joint solution of Equations (6) and (7) determines the coordinate of the break point as 7.3 mM. This is the CMC. The CMC values of SDS known in the literature lie in the range of 8.0–8.4 mM [18], so the value obtained above is slightly lower. However, this is not surprising. It is well known in colloid science that when using solubilization data to find CMC, underestimates are always obtained [1]. These are true CMCs, however, because the presence of a solubilizate changes the nature of a micelle and its properties. It was noted in Ref. [15] that the CMC value of SDS corresponding to the onset of an increase in the NR fluorescence intensity is in the concentration range of 6–8 mM. Finally, it should be noted that in a rigorous thermodynamic theory, the effect of solubilization always reduces to a decrease in the CMC of a surfactant both in the case of saturated solubilizate solutions and in the case of solutions of a given concentration [19].

### 3.2. Extinction Coefficient and Solubilization Curve

To go from the function shown in Figure 4, to a function called the solubilization curve, one needs to know the extinction coefficient *ε* for NR in aqueous solutions of SDS. The extinction coefficient plays the role of the coefficient of proportionality between the optical density (extinction) *A* and the concentration of the absorbing substance in solution *c* for a given optical path length *l* in the Bouguer–Lambert–Beer law
*A = **ε**cl.*(8)

Naturally, all dissolved particles must be the same and not be subject to changes in the course of the experiment. The simultaneous presence of monomers and dimers does not meet the requirements of this law. However, it is acceptable because it has nothing to do with equilibrium. Since we have enough non-equilibrium spectra with NR monomers alone, we can use them to find the monomer extinction coefficient. To find *ε*, special experiments were carried out. Obviously, the values of *ε* should be different for premicellar and micellar SDS solutions. Let us turn first to micellar solutions.

The experiment was organized as follows. In a micellar solution of SDS at a concentration of $c_{1}=30$ mM, solid NR was dissolved to a concentration of 46 μM. From it (by dilution with the same SDS stock solution at a concentration of 30 mM), NR solutions were prepared in the concentration range 1–18 μM, each with the SDS concentration still 30 mM (the change in the volume of the solution during the dissolution of solid NR was neglected). The absorption spectra of these solutions (Figure 5) are unchanged in both shape and position of the maximum (λ_max_ = 579 nm), which is characteristic of the NR monomeric state. The experimental values of the optical density (*A*) at the maxima form the calibration curve (Figure 6), which is easily approximated by a linear dependence A(c_2_) with a slope angle of 3.4 × 10^4^ M^–1^. If we take into account that the experiments were carried out in cuvettes with an optical path length of 1 cm, then for the monomer extinction coefficient NR in the SDS micellar medium, we obtain the value $\varepsilon =3.4\times 10^{4}\;M^{-1}{\mathrm{cm}}^{-1}.$

Turning to the premicellar region, we do not refuse the temptation to test the value of the extinction coefficient found above to calculate the concentration of NR monomers in media with SDS below the CMC. At $c_{1}=$1.5 mM, we have *A* = 0.0078 (curve *1* in Figure 3a). Then, at ε = 3.4 × 10^4^ M^−1^cm^−1^, we obtain $c_{2}=$ 0.23 μM. This is less than the solubility of NR even in water without SDS and is therefore completely unrealistic. Indeed, NR in a premicellar SDS solution should have its own extinction coefficient.

To find it, we used the spectrum of solutions with low concentrations of NR $\left(c_{2}\right)$ and SDS $\left(c_{1}\right)$ in the presence of only NR monomers (Figure 7). Here, the maximum corresponds to the value *A* = 0.0175. Substituting it into Equation (8), we find $\varepsilon =1.25\times 10^{4}\;M^{-1}{\mathrm{cm}}^{-1}$, and for the first four points in Figure 4 we get the values in mM: 0.0006216, 0.0008, 0.001112, and 0.002024. Since the extinction coefficient for the micellar area was found above, we can now go from the dependence $A(c_{1})$ in Figure 4 to the solubilization curve $c_{2}(c_{1})$ (Figure 8, both the concentrations $c_{1}$ and $c_{2}$ are expressed in mM).

Like the dependence $A(c_{1}),$ the solubilization curve consists of two sections (1 and 2, it can be said that the first corresponds to protomicelles, and the second to micelles) separated by the CMC. If the sections are approximated by straight lines, then the CMC can be found by their intersection. If the entire solubilization curve is constructed as a continuous line (it is visible in Figure 8 that the fourth point of Section 1 already lies above the straight line), then the CMC is defined as the point of maximum curvature of the line [1]. Recall that in classical colloid science, Section 1 is generally ignored, and Section 2 is called the solubilization curve, while the CMC is determined at the intersection of straight line 2 with the abscissa axis. Our method with two straight lines is more accurate and, as can be seen from Figure 8 gives the CMC value always slightly higher than the classical method.

Let us perform this procedure. Fitting curves *1* and *2* with straight lines gives, respectively (both concentrations in mM):(9)$$c_{2}=0.0002c_{1}+0.0001\;(R^{2}=0.8966,\;c_{1}\;<\;\mathrm{CMC})$$
(10)$$c_{2}=0.0044c_{1}-0.0325\;(R^{2}=0.9988,\;c_{1}\;>\;\mathrm{CMC})$$

The intersection of the straight line corresponding to Equation (10) with the abscissa axis $\left(c_{2}=0\right)$ gives the SDS CMC value in the presence of NR equal to 7.4 mM. The intersection of the straight lines corresponding to Equations (9) and (10) leads to a value of 7.8 mM. Recall that we obtained 7.3 mM from Equations (6) and (7). All these values are of the same order, but it should still be noted that the SDS CMC value obtained from Equations (6) and (7) is closer to experiment and is not subject to errors in the calculations of the extinction coefficient.

In connection with the construction of the solubilization curve in the premicellar region, it is appropriate to make the following remark. Of the above four values of the points of this curve, the first three lie almost perfectly on the same straight line, and the addition of the fourth point significantly worsens the R^2^ indicator (the linear correlation coefficient, see Figure 9). This is in full agreement with Equation (4), which predicts a linear dependence in the limit of low concentrations. Obviously, the fourth point already requires taking into account the subsequent terms of the expansion of the exponent in Equation (4). It can be said, therefore, that the experiment confirms Equation (4).

Let us now turn to the traditional second section of the solubilization curve. In accordance with the definition expressed in Equation (1), the coefficient at $c_{1}$ in Equation (10) is equal to the solubilization capacity of SDS micelles: *s* = 0.0044. Assuming *z* = 1, we find the SDS aggregation number n≈227. In ordinary “empty” SDS micelles, the aggregation number is theoretically (with perfect packing) 55.5, and in practice it is about 70 [1], so we have a triple excess of this number. 

To verify this fact, let us turn to solubilization theory (see Chapter 7 in Ref. [1]). The effect of increasing the aggregation number under the influence of solubilizate is given by the formula (Equation (52.15) in Ref. [1]).
(11)$$\frac{n}{n^{\prime }}={\left(1+\emptyset \right)}^{2}$$
where *n* is the increased aggregation number, $n^{\prime }$ is the aggregation number in the absence of solubilizate in the micelle, and ∅ is the ratio of the volumes of solubilizate and surfactant in the hydrocarbon core of the micelle. In our case, the volume of one NR molecule is 0.44 nm^3^, and the volume of one hydrocarbon tail of the SDS molecule is 0.3233 nm^3^, so that ∅ = 0.44/0.3233 *n*. Substituting this into Equation (11) and setting $n^{\prime }$ = 70, we arrive at an equation for *n*:(12)$$\frac{n}{70}={\left(1+\frac{0.44}{0.3233n}\right)}^{2}$$

From here, we immediately find the solution *n* = 72.647, which means that when one NR molecule enters the SDS micelle, the aggregation number increases by about 4%, but by no means three times. In other words, the result obtained above for *z* = 1 is completely unrealistic. There can be only one reason for this: the unreality of the condition *z* = 1 itself. Indeed, in an equilibrium micellar system, there are a number of statistical distributions, including the solubilizate distribution. Some micelles contain the solubilizate, while others do not, so the average *z*-number should be less than one. According to Equation (1) *z = sn*, and at *s* = 0.0044 we get *z* = 0.308 for *n* = 70 and *z* = 0.320 for *n* = 72.647. Within these values is the true value of *z*.

It is of note that the coefficient at $c_{1}$ in Equation (9) cannot by analogy be called the solubilization capacity of protomicelles, since they are still being formed as NR nano-adsorbent particles, on which SDS is adsorbed. This coefficient characterizes the solubilization capacity of the entire solution, showing that as the concentration of SDS increases, so does the concentration of NR, but ten thousand times slower according to Equation (9).

### 3.3. SDS in Aqueous Solutions of NR of a Given Concentration

Now there is no precipitate, and all the amount of NR is in the dissolved state. Let us see how the system behaves in this case. To begin with, it is important to dwell on the preparation of the initial solution. A micellar solution with total concentrations of SDS and NR $c_{1}=$ 33.6 mM and $c_{2}=$ 46 μM, respectively, in which complete dissolution of the dye was ensured, was diluted with distilled water approximately 15 times to concentrations $c_{1}=$ 2.2 mM and $c_{2}=$ 3.03 μM. Such a transition through the CMC to the premicellar region is fraught with changes in state. First, all micelles decomposed into monomeric SDS ions in solution and protomicelles, the nuclei of which (NR molecules as nano-adsorbents) are only slightly covered with SDS ions. Such nuclei with an exposed hydrophobic surface are prone to aggregation. Second, NR molecules, remaining in solution (crystallization centers are needed for precipitation), pass into a supersaturated state. To estimate the degree of supersaturation, it suffices to extend curve *1* in Figure 8 to a concentration of 33.6 mM and compare the ordinates of curves *1* and *2*—obviously, these will be not just times, but several tens of times. The situation here is the same as when adding a small portion of a concentrated solution of NR in methanol to water (essentially, diluting this portion with a large amount of water) [7]. Referring to this work, we can expect intense formation of NR dimers in the initial solution.

The absorption spectra of the initial solution in Figure 10 lives up to this expectation. They immediately show maxima related to dimers (λ_max_ = 523 nm) and monomers (λ_max_ = 570 nm) [9,20]. Moreover, from a comparison of the spectra after 1.5 h and 6 days, it can be seen that in the future aggregates of higher orders are formed at the expense of NR monomers and dimers (their peaks decrease with time). This is all that can be said about the initial solution with concentrations of $c_{1}=$ 2.2 mM and $c_{2}=$ 3.03 μM. After reaching its equilibrium state, other working solutions were prepared by gradually adding (as a dry powder) new portions of SDS. In this case, due to the smallness of all concentrations, the change in the volume of the solution, as well as the change in the concentration $c_{2}$, was neglected. However, if we adhere to absolute rigor, this study was carried out not at a constant concentration but with an unchanged amount of NR in the system.

The resulting spectral data are shown in Figure 11. As follows from Figure 11a, at SDS concentrations in the range 5.5 > $c_{1}$ > 2.2 mM (curves *1*–*3*), the optical density increases, which indicates the reverse process of the formation of NR monomers and dimers (λ_max_ = 572 nm and λ_max_ = 520 nm, respectively) due to the decomposition of higher-order aggregates with a predominance of dimers. At $c_{1}=$ 7.5 mM (curve *4*), the absorption band characteristic of NR dimers becomes less pronounced, and monomers come to the fore. At concentrations $c_{1}=$ 9 mM, which exceeds the SDS CMC value, the absorption spectrum (curve *5*) with a clearly defined maximum at 579 nm indicates the presence of mainly NR monomers. A further increase in the SDS concentration leads to a slight increase in the optical density (curves *6* and *7* in Figure 11b), the growth of which stops at $c_{1}\approx$10 mM.

To explain the observed changes in the absorption spectra of NR solutions, we consider that, in the premicellar region of the SDS, the dye is in the form of a protomicelle with a nucleus containing mainly its dimers. Above the CMC, protomicelles with a monomeric nucleus can also form through ordinary micelles by inclusion in them of solubilizate molecules. Therefore, starting with the concentration of $c_{1}\approx$ 10 mM, in the solution of the SDS there are mostly monomers of the dye in the form of solubilization nuclei of the micelle. This can be judged by the absence of changes in the absorption spectra of NR with a further increase in the concentration of SDS.

The maxima of monomers and dimers in the spectra of Figure 11 are quite close and may partially overlap each other. In such cases, to clarify the picture, a special operation of spectrum decomposition is usually used, for which special programs have been created. We had the PeakFit mathematical package (Version 4 for Win 32, AISN Software Inc., Mapleton, OR, USA) at our disposal, and we applied it to the spectra of Figure 11. Figure 12 illustrates how this is done using the example of the decomposition of curve *1* in Figure 11. This refines the height and location of the peaks in the spectrum. The results obtained are summarized in Table 1 and shown in Figure 13.

It can be seen from Figure 13 that the optical density, and, consequently, the concentration of dimers also passes through a maximum in the premicellar region. This was justified and confirmed theoretically in Section 3.1.

Since the optical density is proportional to the concentration of dissolved substances, according to Figure 12, the change in the concentration of monomers and dimers can also be judged. Curve *1* shows that the concentration of dimers passes through a maximum inside the premicellar region. This was already encountered in the previous section and gave this phenomenon a justification based on the mass action law. Since the mass action law is universal and, obviously, also operates in the case under consideration, we can only say that the shape of the curve for dimers in Figure 12 confirms the predictions of the theory. In this case, the dimers continue to exist almost until the CMC, after which they disappear. In other words, NR monomerization under the influence of SDS occurs no earlier than when the CMC value is reached for SDS. NR significantly differs from phthalocyanines, for which monomerization was achieved before CMC [3,4,13].

## 4. Conclusions

Summing up, there are a number of original aspects of this work to be noted. The first is that equilibrium data are presented for the first time for the system under consideration. They relate to two processes occurring in time, the dissolution of NR in water or aqueous solutions of SDS and the establishment of aggregation equilibria. Due to the very low solubility of NR in water (about 2 μM), reaching a saturated state of the solution requires several weeks. If SDS is present in the solution, the process is faster. It should be kept in mind that the mechanism of dissolution is molecular, i.e., individual NR monomers pass into the dissolved state. As they accumulate, the process of aggregation develops, starting with the formation of dimers. All these processes are easily followed using absorption spectra. The achievement of an equilibrium state is fixed from the moment of cessation of any spectral changes. The NR spectrum shows the presence of only monomers in two cases: (1) if the state is nonequilibrium and dimers have not yet appeared; (2) if equilibrium is reached but the NR concentration is too low to detect dimers. In both cases, the monomer extinction coefficient for NR can be easily found using the Bouguer–Lambert–Beer law, Equation (8) (which we used), although this law does not work in the presence of dimers.

The second original aspect is the study of NR solubilization not only in micellar, but also in premicellar SDS solutions. In the latter case, solubilization is achieved due to the adsorption activity (as a result of the hydrophobic effect) of SDS ions on the surface of NR molecules, which play the role of a nano-adsorbent, with the formation of protomicelles. Accounting for this phenomenon leads to a refinement of the method for finding the CMC from the solubilization curve as the point of intersection of its branches corresponds to micelles and protomicelles. From the classical micellar branch of the solubilization curve (more precisely, from its straight section), the SDS aggregation number in a micelle with one solubilized NR molecule was found to be 227. This is three times higher than in the “empty” SDS micelle in water, whereas theory predicts only an increase of a few percent. The situation is that some micelles remain empty, so that the average number of molecules of NR per one micelle is smaller than unity (*z* < 1) in reality. As for the premicellar branch of the solubilization curve, it has been shown theoretically and verified experimentally that, at the limit of low concentrations, the dependence is linear. The general regularity of the solubilization curve, $dc_{2}/dc_{1}>0,$ was also established theoretically. This was also confirmed experimentally.

As a third aspect, we note a detailed study of the molecular aggregation of NR in premicellar aqueous solutions of SDS. For dyes, this problem is also of great practical importance, since only dye monomers have chromophoric properties. The study was carried out both in the process of dissolving NR and when diluting the micellar solution of NR with water and returning to the initial state by adding SDS. Especially indicative was the second option. When a micellar solution is diluted to an SDS concentration below the CMC, all the given amount of NR is in the dissolved state, but at a concentration much higher than the solubility. In other words, we have a supersaturated NR solution, and supersaturated solutions, as is known, form metastable states (equilibrium states stable with respect to infinitesimal changes), in which they can exist for quite a long time in the absence of crystallization centers. Once in water, the monomeric NR molecules immediately stick together forming dimers under the influence of the hydrophobic effect, but it also affects the dimers and all subsequent degrees of aggregation. The aggregation process develops, the dimers and monomers stick together into trimers, and the dimers themselves into tetramers, etc. This process is fixed by the absorption spectra, in which the height of the maxima of monomers and dimers decreases when aggregates of higher orders are formed due to them. Ultimately, in a supersaturated solution, an equilibrium size distribution of molecular aggregates is established with the predominance of dimers.

If we now gradually add SDS to the system, the opposite picture of the destruction of NR aggregates will be observed. However it should not be thought that they are destroyed under the impact of SDS ions. For a correct understanding of the whole picture, it should be remembered that we are dealing with a dynamic system. Under the action of thermal motion, each aggregate has its own lifetime (on average, for example, 10^–3^–10 s for micelles, and the residence time of one ion in a micelle is of the order of 10^−7^–10^−6^ s [21]). Aggregates themselves are destroyed and re-created. However, if the concentration of ions in solution is high enough, they can interfere, recreating aggregates. This is their destructive power. At the same time, all aggregates play the role of nano-adsorbents for SDS, and hence the potential creators of protomicelles. Not only monomers, but also dimers and larger aggregates can serve as solubilization cores of protomicelles. Therefore, at the first stage, after the addition of SDS, we have the same aggregates partially covered with adsorbed SDS ions. As the concentration of SDS increases, the proportion of the surface occupied by SDS ions also increases. Therefore, they begin to occupy the surface that appears during the decomposition of aggregates, reducing the likelihood of subsequent sticking together. The number of aggregates decreases, and the final state is a micellar system with solubilized monomers. This is the mechanism of dye monomerization under the action of surfactants. In this case, a micelle with a solubilizate can arise independently from both protomicelles and ordinary micelles. In the case of phthalocyanines, we observed dye monomerization through protomicelles below the CMC, while in the case of NR, both mechanisms are implemented almost simultaneously.

## Figures and Tables

**Figure 1 molecules-27-07667-f001:**
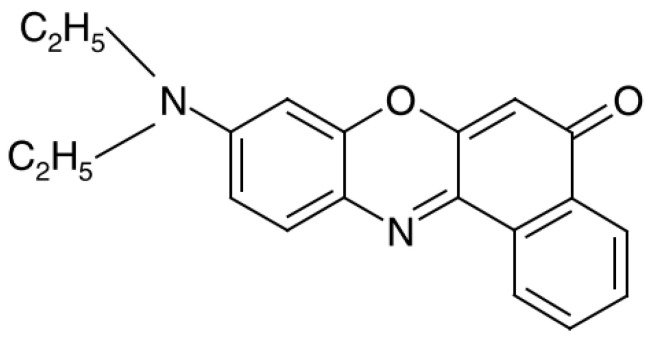
The structural formula of Nile red (NR).

**Figure 2 molecules-27-07667-f002:**
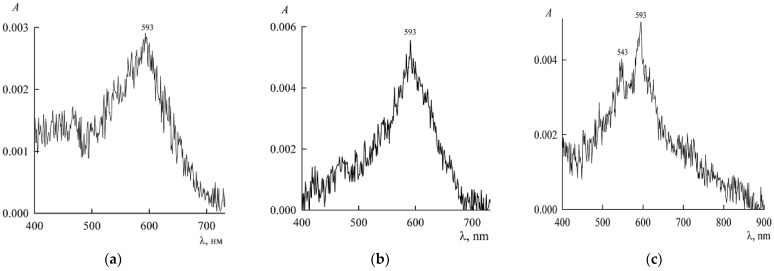
Absorption spectra of NR in a dispersion with water with a gross concentration of 2.5 μM after 1 day (**a**), 7 days (**b**), and 16 days (**c**) after preparation (*A* is optical density, λ light wavelength).

**Figure 3 molecules-27-07667-f003:**
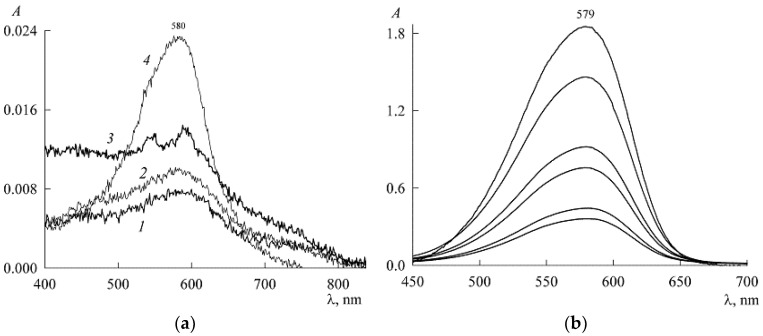
Absorption spectra of saturated NR solutions (with a content of 440 μM in the dispersion) in the presence of SDS at concentrations $c_{1}$, mM: (**a**) 1.5 (1), 3 (2), 5.2 (3), and 7.2 (4); (**b**) from bottom to top: 9.1, 10.3, 12.3, 14.4, 19.4, and 21.

**Figure 4 molecules-27-07667-f004:**
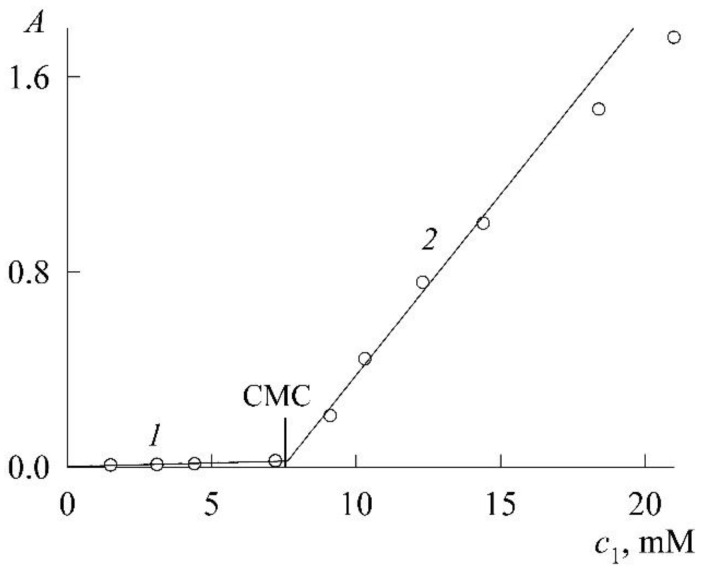
Dependence of the optical density (*A*) of saturated NR solutions on the SDS concentration (*c*_1_).

**Figure 5 molecules-27-07667-f005:**
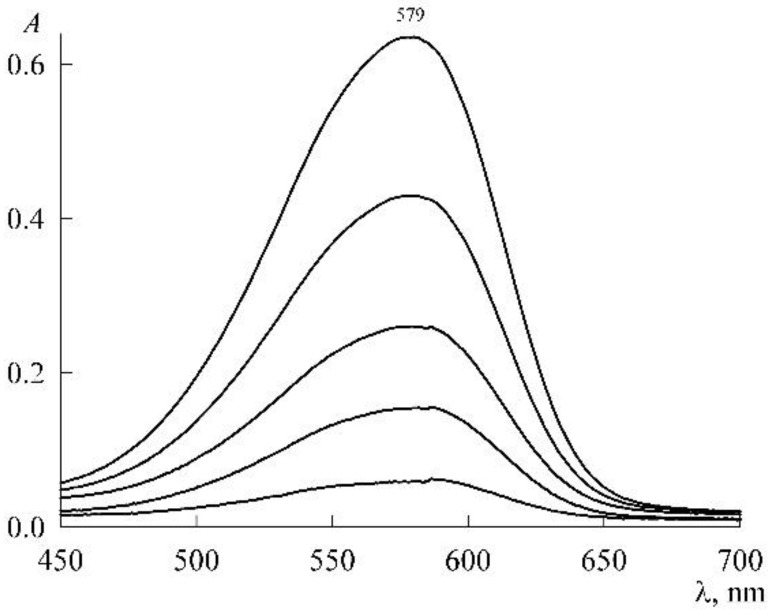
Electronic absorption spectra of NR solutions in the presence of 30 mM SDS with NR concentrations (from bottom to top, μM) 1.38, 4.2, 7.1, 12, and 18.

**Figure 6 molecules-27-07667-f006:**
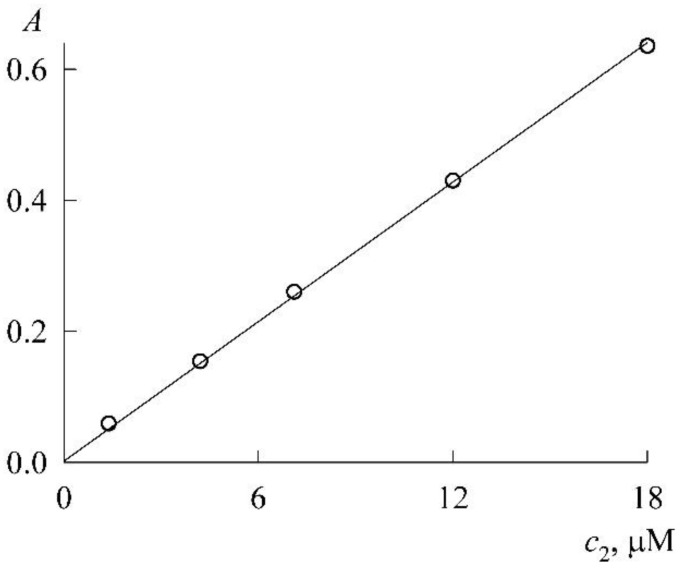
Calibration curve for NR in 30 mM SDS solution at the wavelength of 579 nm.

**Figure 7 molecules-27-07667-f007:**
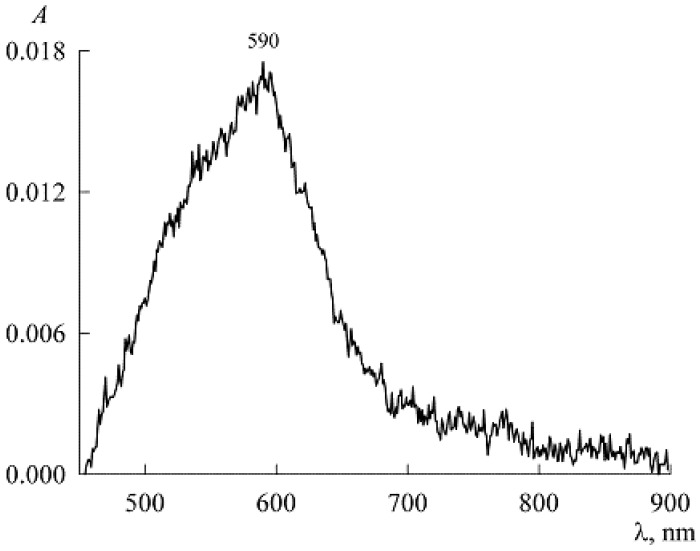
Absorption spectrum of NR in a weak solution of SDS at concentrations *c*_1_ = 1.5 mM and *c*_2_ = 1.4 μM.

**Figure 8 molecules-27-07667-f008:**
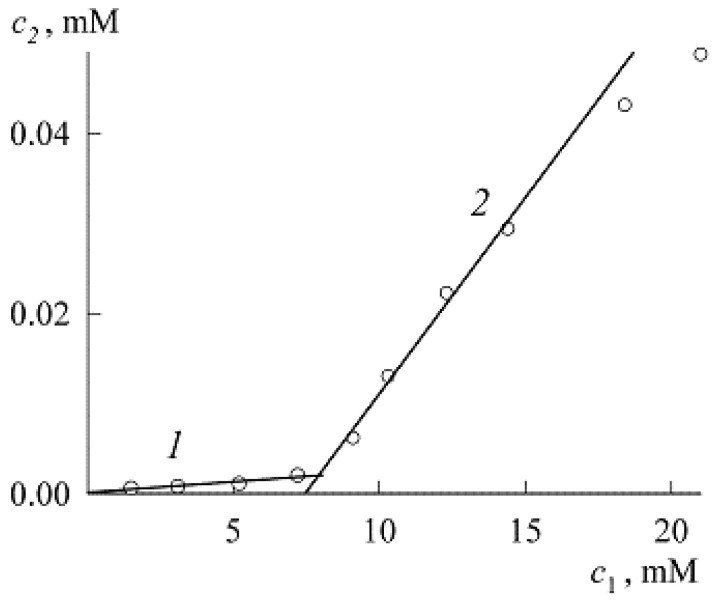
Solubilization curve for NR in premicellar solution (Section 1) and in SDS micelles (Section 2).

**Figure 9 molecules-27-07667-f009:**
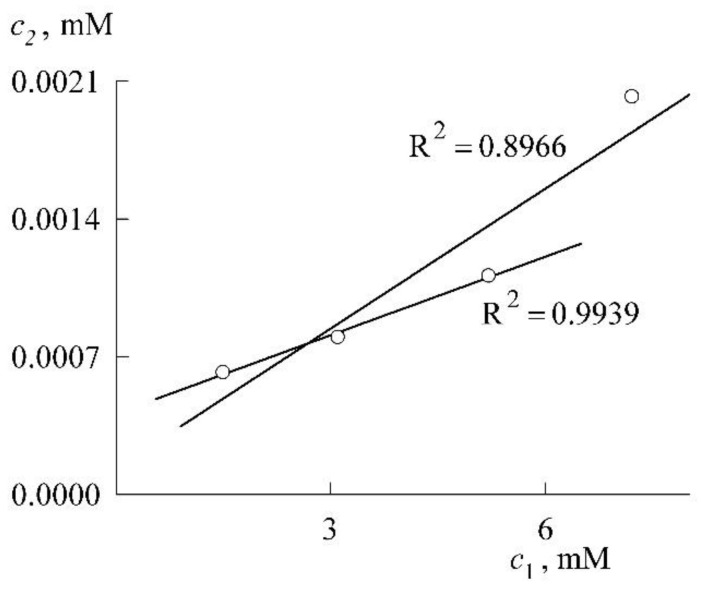
Verification of Equation (4).

**Figure 10 molecules-27-07667-f010:**
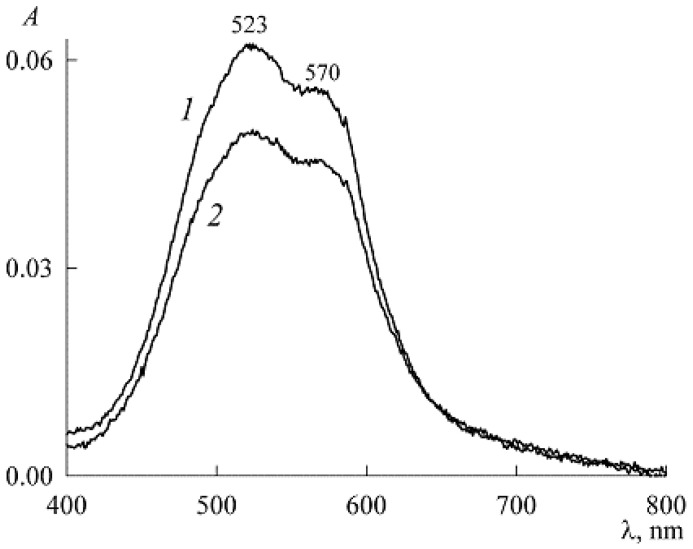
Absorption spectra of the stock NR solution (3.03 μM) in the presence of 2.2 mM SDS, 1.5 h (curve *1*) and 6 days (curve *2*) after preparation.

**Figure 11 molecules-27-07667-f011:**
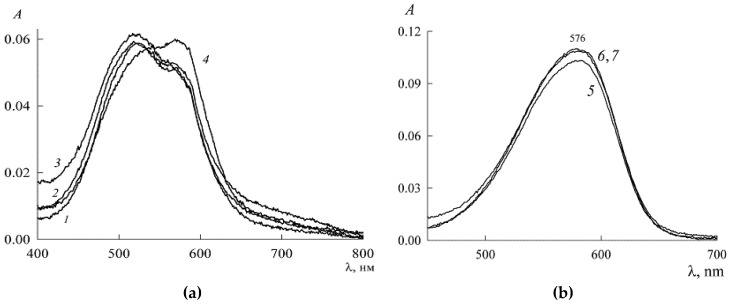
Absorption spectra of SDS solutions in the presence of NR ($c_{2}=$ 3.03 μM) at values of $c_{1}$, mM: (**a**) 2.2 (1), 2.8 (2), 5.5 (3), and 7.5 (4); (**b**) 9.1 (5), 10 (6), and 10.5 (7). Optical path length *l* = 1 cm.

**Figure 12 molecules-27-07667-f012:**
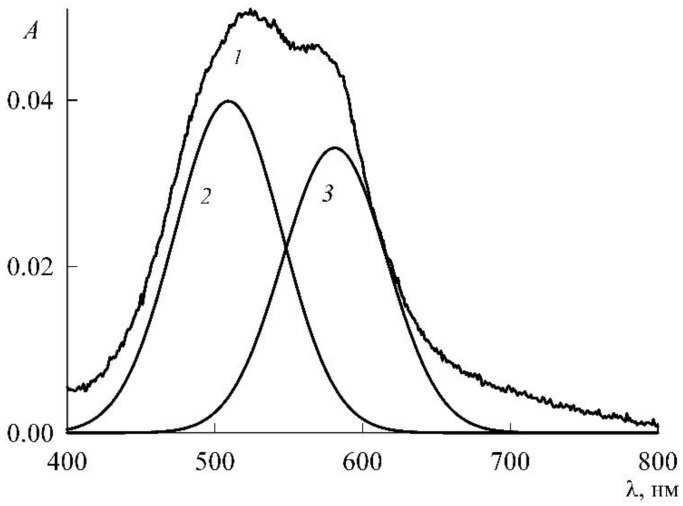
An example of decomposition of Curve *1* on Figure 11 in two components (*2* and *3*).

**Figure 13 molecules-27-07667-f013:**
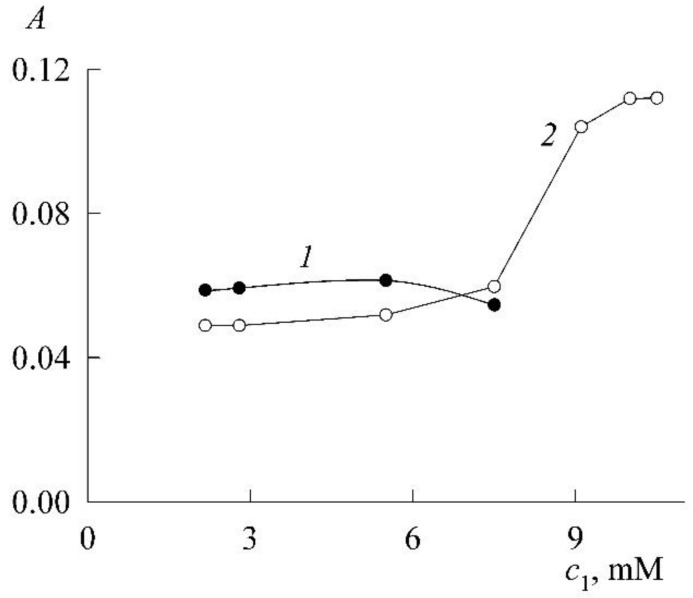
Function $A(c_{1})$ for dimers (curve *1*) and monomers (curve *2*) of NR in the SDS solution at the NR concentration of 3.03 μM, plotted from Table 1 and the maxima of the spectra in Figure 11 at a wavelength near 520 ± 2 nm for dimers (curve *1*) and in the range of 570 ± 2 nm for monomers (curve *2*) of NR.

**Table 1 molecules-27-07667-t001:** The results of decomposition of the spectra in Figure 11 for the monomeric and dimeric components of NR in the premicellar region of SDS solutions.

Curve Number	*A* for λ_max_	
	For Monomers	For Dimers
1	0.039	0.043
2	0.0419	0.0499
3	0.0414	0.05
4	0.0496	0.0416

## Data Availability

Not applicable.

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
