# Peer review of "Solubilization of Nile Red in Micelles and Protomicelles of Sodium Dodecyl Sulfate"

_molecules, 2022, doi:10.3390/molecules27227667_

Round 1

Reviewer 1 Report

This work presents new and interesting data for a well known system. Nile Red in SDS micelles. I attach a pdf annotated file with my analysis of this work. My main critics are:

- referring to SDS as an ion

- drawing conclusions on small variations of Absorbance not properly making a spectral decomposition of the various components that might contribute (monomers, dimmers, higher order agreggates)

- some reasoning is presented at a very low level while some other aspects just pop out without justification: "The extinction coefficient plays the role of the coefficient of proportionality between the optical density (extinction) A and the concentration of the absorbing substance in solution" vs " But now we are talking about SDS micelles with a large solubilization core (NR molecule), and this makes the aggregation number 227 quite realistic" (see my comment on this sentence)

The writing is somewhat as if the authors were speaking.

After correction this work might be considered for publication.

Author Response

Please see attached pdf file.

Reviewer 2 Report

In his 89th year of life, Professor Rusanov worked out the thermodynamics of nano-adsorbents (large hydrophobic molecules) and the coating of nano-adsorbents with surfactants - the formation of protomicelles (Colloids Surf. A. 2021, 629, 127453). In this manuscript, Rusanov and colleagues (Movchan and Plotnikova) continue to develop the theory (through experiments where the nano-adsorbent is Nile Red, while the surfactant is Sodium Dodecyl Sulfate (SDS)) of the nano-adsorbent, the monomerization of Nile Red with SDS...

Manuscript is very concisely written experimental results and discussion match well, there is enough experimental evidence in the form of Figures.

The conclusion can be a little long, which could be included in the discussion and only highlights can be given in the conclusion, but it can also remain in this form.

Minor:

1. The authors on page 2 (line 63) use the term hydrophobic interaction while on page 5 (line 175) they use the term hydrophobic effect. Do the authors use these terms as synonyms since usually the hydrophobic effect refers to the departure of water molecules from the hydration layer above the hydrophobic surface into the interior of the solution (bulk) - entropic contribution; while hydrophobic interactions are intermolecular interactions between hydrophobic surfaces - an enthalpic effect that is significant at (relatively high) temperatures when the entropic contribution of the hydrophobic effect is zero.

2. When surfactants are adsorbed on a nanoadsorbent, does their surface tension profile (solution/air) change depending on the concentration compared to the surface tension profile without the nanoadsorbent.

3. When SDS coats monomeric NP particles, why does it not start self-association (micelle formation at lower critical micellar concentrations) but affects the separation of dimeric NP particles and the formation of new monomers, how can this phenomenon be thermodynamically described, i.e. why it is more favorable than self-association of SDS anions.

Author Response

Please see attached pdf file.

Reviewer 3 Report

In the present manuscript, authors have reported the solubilization and aggregation of the nile red dye (NR) in premicellar and micellar aqueous solutions of sodium dodecyl sulfate (SDS). The subject is interesting however there is need to improve the English of manuscript and also there are certain quarries which must be satisfied before final publication:

 1) In line 61, authors state “according to Ref. 6, less than 1 g/mL…..” 1 g/mL is an appreciable solubility so this needs to be confirmed.

2) In line 92, correction is needed in the units of electrical conductivity as “mS / m”.

3) In line 94, authors mentioned two values for optical path lengths of quartz cuvettes i.e 0.1 and 1 cm. For which experiments 0.1 cm quartz cuvette is used? Please mention at appropriate place in manuscript. Also in the sentence it should be “lengths” if two different cuvettes were used.

4) What was the pH of the aqueous solution? Is there any chance of protonation of terminal –N- of NR dye in aqueous solution? In such case locus of solubilization of NR will not be core of micelle. Discuss this aspect in revised manuscript.

5) In line 172-173, the sentence “We have such power” is not clearly explaining the point. This sentence needs to be revised.

6) What is “KKM” in equations 6 & 7 (page 7)? It is not defined in text.

7) At many places e.g lines 222 & 227 authors have stated “The SDS CMC values…” this should be replaced with “The CMC Values of SDS….”

8) Line 246, why 30 mM solution of SDS was used for post micellar concentrations as this concentration is about 4 times higher than the CMC of SDS which is around 8 mM. Shape of micelles depends on concentration of surfactant. What about the shape of SDS micelles at such high concentration of SDS. Will this affect the present study?

9) Figure 7 is missing in the manuscript.

10) Paragraph from line 268 to 275 is not clear, may be due to missing Figure 7. Make it clear

11) On What basis, z (the number of solubilizate molecules in a micelle)  is assumed as 1 in line 310.

Author Response

Please see attached pdf file.

Round 2

Reviewer 1 Report

The authors have complied to all my comments/corrections.

A small error was now introduced. SDS stands for Sodium Dodecyl Sulphate. If the surfactant is ionized it should be DS- and not DS2- as used in the text.

After this minor correction this manuscript can be accepted for publication in Molecules.